# Mitigation Planning and Policies Informed by COVID-19 Modeling: A Framework and Case Study of the State of Hawaii

**DOI:** 10.3390/ijerph19106119

**Published:** 2022-05-18

**Authors:** Thomas H. Lee, Bobby Do, Levi Dantzinger, Joshua Holmes, Monique Chyba, Steven Hankins, Edward Mersereau, Kenneth Hara, Victoria Y. Fan

**Affiliations:** 1Thompson School of Social Work & Public Health, University of Hawaii at Manoa, Honolulu, HI 96822, USA; tlee@hawaiidata.org (T.H.L.); bdo7@hawaii.edu (B.D.); levidantzinger@gmail.com (L.D.); jrholmes@hawaii.edu (J.H.); 2Hawaii Data Collaborative, Honolulu, HI 96813, USA; 3Department of Mathematics, College of Natural Sciences, University of Hawaii at Manoa, Honolulu, HI 96822, USA; chyba@hawaii.edu; 4John A. Burns School of Medicine, University of Hawaii at Manoa, Honolulu, HI 96813, USA; hankinss@hawaii.edu; 5Behavioral Health Administration, Hawaii Department of Health, Honolulu, HI 96813, USA; phac@hawaii.edu; 6Hawaii Department of Defense, Honolulu, HI 96816, USA; kenneth.s.hara@hawaii.gov; 7Center for Global Development, Washington, DC 20036, USA

**Keywords:** COVID-19, pandemic, modeling, epidemiology, isolation and quarantine, media and communication, public health planning, governance, hospital, pandemic preparedness

## Abstract

In the face of great uncertainty and a global crisis from COVID-19, mathematical and epidemiologic COVID-19 models proliferated during the pandemic. Yet, many models were not created with the explicit audience of policymakers, the intention of informing specific scenarios, or explicit communication of assumptions, limitations, and complexities. This study presents a case study of the roles, uses, and approaches to COVID-19 modeling and forecasting in one state jurisdiction in the United States. Based on an account of the historical real-world events through lived experiences, we first examine the specific modeling considerations used to inform policy decisions. Then, we review the real-world policy use cases and key decisions that were informed by modeling during the pandemic including the role of modeling in informing planning for hospital capacity, isolation and quarantine facilities, and broad public communication. Key lessons are examined through the real-world application of modeling, noting the importance of locally tailored models, the role of a scientific and technical advisory group, and the challenges of communicating technical considerations to a public audience.

## 1. Introduction

The health and economic toll of the global coronavirus disease 2019 (COVID-19) pandemic posed unprecedented challenges for how public health authorities respond and mitigate a public health emergency and crisis globally. Yet, the local public health response to COVID-19 was challenging for many reasons, including the widespread uncertainty as well as ever-evolving science and knowledge of a new disease, the wide-ranging mitigation measures and health and socioeconomic impacts, the disproportionate impact on vulnerable populations, the highly charged political context and polarized communication challenge, the lack of preparedness and capacity for public sector response, among others [1,2,3]. However, in many countries, COVID-19 was mitigated through local or subnational efforts and responses in addressing the multidimensional impacts of COVID-19. As a result, local policymakers and public health authorities were challenged to make timely decisions and deploy a variety of tools to best respond to the emergency.

In many countries, one key tool used and deployed by policymakers was mathematical modeling and epidemiologic forecasting of infectious diseases. There was widespread use of a variety of models that forecasted and made predictions about the spread of COVID-19 and its subsequent health impacts. The COVID Forecast Hub (https://COVID19forecasthub.org/ (accessed on 11 May 2022)) curated a partial list of more than 40 models, predominantly from universities and research institutes and with a wide range of predictions on case counts, hospitalizations, intensive care unit use, and deaths. The landscape of models can be dizzying for a non-technical user including policymakers.

Policymakers can use mathematical and epidemiologic modeling to inform a variety of pressing policy, programmatic, and planning questions and decisions for mitigating COVID-19. These decisions can be grouped into two main categories along the slopes of the COVID-19 curve—the surge up and the decline down. Key questions for policymakers seeking to make decisions informed by best available science and evidence through modeling include:Given the nature of COVID-19 to surge exponentially, are there sufficient health care resources in our jurisdiction or state, including hospital beds, ventilators, personal protective equipment, medication, isolation and quarantine facilities, contact tracers, and other health workers? Will capacity be sufficient, and if not, when will they run out? Does the state need to enforce stronger mitigation measures including at its extreme “shutdown”, i.e., mass quarantine and isolation for the state?As COVID-19 ostensibly declines, what policy decisions should authorities undertake to reopen and relax measures? What mitigation measures need to be maintained and what measures can be dispensed including testing, tracing, isolation, masking, distancing, and vaccination?

This study presents a real-world historical case study of the roles, uses, and approaches to COVID-19 modeling and forecasting for policy decisions and policy use cases, drawing from the historical perspectives in one state jurisdiction in the United States. This study does not present a micro-level analysis of detailed modeling and its mathematical specifications, but rather provides a macro-level historical and policy perspective on the ways in which modeling informs policymaking. The methodology and data used for this case study rely on a review of the historical facts and real-world events through lived experiences of the authors of this paper who are members of the Hawaii Pandemic Applied Modeling Work Group (HiPAM) (https://www.hipam.org (accessed on 11 May 2022)), the Hawaii Department of Defense, Hawaii Emergency Management Agency (HI-EMA), or the Hawaii Department of Health in a variety of roles during the COVID-19 pandemic from March 2020 to May 2022.

This case study is intended to and may help future policymakers seeking to navigate this complex landscape of models and draw upon practical lessons learned on how to make appropriate evidence-based decisions using models. As such, this paper is structured as follows. The first part of this case study focuses on the major technical considerations of mathematical and epidemiologic models and how models were selected given real-world limitations of time and resources. The second part of this case study reviews and summarizes the real-world policy use cases and key policy decisions informed by modeling during the pandemic surge and decline, including the role of modeling in informing planning for hospital capacity and isolation and quarantine facilities, and deploying a broad public communication strategy that navigates the complexities and pitfalls of modeling. We then reflect on the key lessons and discussion from the use cases that may be relevant for other jurisdictions seeking to use modeling to inform decision making.

## 2. COVID-19 Models Used to Inform Policymakers in Hawaii

### 2.1. Model Selection

A “model” refers to a mathematical or logical representation of the biology and epidemiology of disease transmission and its associated processes [4]. To date, there are more than 40 COVID-19 models available. In many locations around the world, there was a need for timely public health decisions pressing against the lack of available time, resources, and a severe dearth of expertise such as epidemiologists in the jurisdiction (as was the case in Hawaii). Thus, it was not feasible nor practical for policymakers and their technical support teams to comprehensively review all models in order to make decisions. Instead, policymakers were continuously forced to make strategic decisions to select and use tools in order to make the best available information at hand. Nevertheless, even the selection of tools in order to make policy decisions requires technical expertise and communication savvy in order to wade through the science and complexity and minimize creation of additional confusion.

The first part of this case study focuses on the major criteria for selecting a model. Beginning in April of 2020, the HI-EMA tasked a technical team (including a physician, a lead epidemiologic adviser and a technical analyst) who in turn sought guidance from a newly developed HiPAM work group to review and use models to inform a variety of specific policy decisions, described in the second part of this case study. The four models that were ultimately chosen and used for informing Hawaii public authorities for decision making were based on selective review rather than comprehensive review of models. The four models used by the technical team were the following: the University of Washington Institute for Health Metrics and Evaluation (IHME) model [5], the Imperial College London model [6], the Epidemic Calculator [7], and the University of Basel model [8]. The dimensions for reviewing and selecting these models are described in Section 2.2.

Upon review of the documentation and source code, if available, of these models, in 2020, the technical team identified some of the key assumptions of these models. These assumptions were crucial in understanding the limitations or applicability of a given model to a particular jurisdiction, and in this case, the state of Hawaii. These assumptions and limitations are discussed in Section 2.3.

### 2.2. Criteria for Model Selection

There are several criteria that could be considered for selecting a model. In this case study, the technical team in Hawaii was prompted with questions from policymakers relying on wide media coverage on two models in particular—the University of Washington and the Imperial College London model. Yet, as the technical team discovered, these two models were not completely suitable or customizable for the situation in the local state jurisdiction. The technical team then identified two more models (Epidemic Calculator and the University of Basel models) and reviewed these four models based on publicly available documentation (noted in the aforementioned references), and in some cases, data visualizations and source code, along five key dimensions. At the time, the COVID-19 modeling hub had not yet been available in the early part of the pandemic, and thus the models chosen were selective and purposive.

The key dimensions used to select and use these four models were the following: (1) model objective, (2) interactivity and local parameter customizability, (3) age distribution, (4) type of model, and (5) open source (see Table 1). Given limitations of time and resources, the technical team made purposive decisions on which models to consider and use in 2020, and compared and contrasted the models along these dimensions. These dimensions were argued to be relevant for decision making in Hawaii based on the issues of the assumptions and limitations of the models. While these are not comprehensive of all considerations, they reflect the historical events in the Hawaii case.

**Model Objective.** Each model had a different objective. The IHME model intended to estimate COVID-19 hospital impacts, whereas the Imperial College London model sought to illustrate how public health measures such as physical distancing and protecting vulnerable populations affected the spread of COVID-19. Understanding the objective of the model is an important but incomplete aspect to its appropriate use.

**Local Parameter Customizability.** Some models allowed for interactivity and customizability of the model parameters. The Epidemic Calculator had sliders to allow for a user to modify parameters driving the transmission and clinical dynamics underpinning the model (e.g., the population size and the basic reproduction number R0) and to add an intervention to decrease transmission by a specified amount from a given day. The Basel model allowed for the user to modify various model parameters, age-group-specific parameters, and isolation measures, and add multiple interventions to reduce transmission. In contrast, the IHME model had limited local parameter customizability. Although it generated state-specific estimates, it did not allow for state-specific parameters to be incorporated. Moreover, although the IHME model used a wide variety of data sources, not all states had their data reflected in the model. In the case of Hawaii, the IHME model initially did not appear to utilize data from Hawaii but instead utilized average estimates of time from hospitalization to death from other states, despite widely different demographic, epidemiologic, and socioeconomic considerations.

**Age Distribution.** Age is well documented as one of the largest and most significant risk factors for COVID-19, with older adults at increased risk of being hospitalized and dying due to COVID-19 [9,10]. Each state has different age distributions and demographic and population age structure, and so it is important for models to account for age to project the case, hospitalization, and fatality numbers more accurately. The technical team during their rapid review identified the University of Basel model as Susceptible Infected Recovered/Susceptible Exposed Infected Recovered (SEIR) compartment-based models that accounted for age distributions, allowing for the user to adjust the age distribution and age-group-specific parameters to reflect the population of interest.

**Type of Model.** Models can be broadly categorized into two types—mechanistic and statistical. Mechanistic models make assumptions about how the actual process of COVID-19 disease transmission occurs and include the SEIR compartmental models and their modified variants. In contrast, statistical models fit curves using existing data, the main example being the IHME model which early on used the existing data from China and Italy to predict what would happen in the United States and elsewhere. This means that while statistical models can forecast what will happen in the near future, mechanistic models can make assumptions on the transmission dynamics of COVID-19 and forecast longer-term scenarios based on different interventions and policy changes [11].

Incorrectly utilizing a statistical model to create long-term scenarios can produce results that “may suffer from the fallacy of Farr’s law, a similar non-mechanistic method in which epidemics are assumed to follow a normal distribution shifted and scaled to fit data” [12]. This was a common and widespread criticism of the IHME, as simply fitting a curve to historical data and extrapolating into the future can produce dramatic over- or underestimates of the epidemic’s impact [13].

However, mechanistic models also have shortcomings. Parameters available to a model are finite, meaning any output will be inherently flawed. In addition, assignment of values to the parameters available in each model may only be viable with respect to a given historical situation, but relatively meaningless considering even a small shift in the makeup or habit of the population or any immediate major policy change in the future, including upon seeing the results of the model. Therefore, it is important to recognize and establish degrees of uncertainty within the parameters themselves for mechanistic models. Models without such extended boundaries should be caveated generously or avoided completely.

**Open Source.** Models that are open source, defined as having the source code made publicly available for use and modification, are models that enable users to “open up the hood of the car” or “look into the sausage-making machine”. This transparency in model assumptions and limitations should have appropriate interpretation by an epidemiologist to policymakers to ensure appropriate planning. Most importantly, open source incurs little to no additional cost and offers support to states with limited technical and epidemiologic capacity. For example, the IHME model was not open source which made it very challenging to assess even basic assumptions such as how it incorporates age-specific distributions. Policymakers should approach model interpretations cautiously and not make assumptions of the data.

### 2.3. Model Assumptions and Limitations

This section reviews the key assumptions identified by the technical team of these models, which were used to inform the applicability of a given model to a particular jurisdiction, and in this case, the state of Hawaii. Table 2 presents explicitly some of the assumptions in the four models. Ultimately, the technical team chose to use the Epidemic Calculator and University of Basel model for several key decisions in 2020, as described in Section 3. In Hawaii in 2020, the Hawaii Data Collaborative in partnership with local hospitals and HiPAM built on and modified the open-source Epidemic Calculator model to show how policy measures on reopening and resuming travel could impact the spread of COVID-19. Later beginning in 2021, HiPAM relied on a locally customized model.

## 3. Policy Use Cases of Applying Models to Specific State Policy Decisions

In this section of the policy case study, we provide a historical account for how models were used to inform key policy decisions in the jurisdiction, both in terms of managing capacity during a surge and reopening amidst a decline. The policy case study reflects actual lived experiences along with (now historical) observations and perspectives of the technical team and the HiPAM work group who were providing information to policymakers seeking to make decisions informed by modeling. Thus, the detailed micro-level analyses for each use case are not presented herein, but rather, the specific translation of evidence to knowledge and communication with a variety of stakeholders including policymakers, media, and the public are described.

### 3.1. Using Models for Managing Resources and Capacity in a Surge

Many states did not have an established epidemic or pandemic response plan for COVID-19, let alone a plan for how to use modeling for informing policy decisions. Amidst this context, a pressing and overarching question was how quickly COVID-19 would spread in their state or community. As such, the IHME model was utilized because it provided early state-specific estimates. It was one of the only models at the time that gave a hard deadline by which a state’s bed surge capacity might be reached because of the speed by which COVID-19 spreads and leads to hospitalization. It was also widely disseminated in the news and prompted several policymakers to inquire whether decisions could be made based on what was circulated in the media. Policymakers rarely have a background in infectious disease or epidemiology, and the wide coverage of the models in media does not guarantee their appropriate use. Thus, this case study reflects the occasion in which some policymakers had the foresight and humility to seek out information and inputs from technical experts for three use cases described herein:**Use Case 1:** Determining whether there was adequate hospital bed capacity in the state and adequate PPE in the state.**Use Case 2:** Assessing the need for isolation and quarantine facilities from the surge of the second wave in the fall of 2020.**Use Case 3:** The role of public communication during the Delta surge in the summer of 2021, and the Omicron surge in the fall of 2021.

#### 3.1.1. Use Case 1: Adequacy of Hospital Bed and Personal Protective Equipment Capacity

The IHME model was initially used to plan for ensuring adequate bed capacity and to decide whether to put up additional acute care facilities. In Hawaii, policymakers pondered challenging decisions of whether to retrofit existing hotel rooms or outfit a convention center. Either option would require collaboration with the US Army Corps of Engineers with an expensive price tag. This policy decision required COVID-19 case and hospitalization projections specific for Hawaii. In the beginning of the pandemic, with no other available guidance or tools as well as limited or no epidemiologic advisors, policymakers turned to the web-accessible IHME model for guidance on when Hawaii would be hit with a “surge” of cases.

However, at the onset of COVID-19 in the US, many states had yet to fully understand how the virus was spreading through their individual communities and how measures such as requiring face mask use in public would affect the spread [14]. Through the month of March and early April, many states did not yet have a high case count and fatality count to get a sense of the trend of COVID-19 within their state. The IHME model used the hospitalization to death ratio from seven locations within the US with the most cases to create a weighted average for their ratio and applied it to states with fewer than five fatalities, which included Hawaii. This resulted in Hawaii expecting to see a surge in cases and hospitalizations that was projected to overwhelm the local healthcare system.

Yet, when the technical team with the HiPAM work group and utilized a basic SEIR model with Hawaii-specific parameters, no surge was estimated within the same time frame that IHME was predicting. The modeling team in Hawaii understood that Hawaii’s unique geography and early mitigation efforts, most significantly restricting air and sea travel, drastically reduced the Rt below the value of 2.2 that was used by most models. The technical team stated all the limitations and assumptions of their early model to the decision makers in HI-EMA and others. Based on the recommendation of state-specific data and use of a locally tailored epidemiologic model, the decision was made to not retrofit the Hawaii Convention Center into an acute care facility at that time, and to re-evaluate at a future date. Ultimately, Hawaii was never hit with a surge at the level predicted by the IHME model, i.e., the predictive validity of the IHME model was poor for Hawaii. Due to the information provided by the technical team, the state of Hawaii made a policy decision that avoided millions of USD in costs.

While the IHME model influenced policymakers and emergency management leaders’ decisions about imposing public health measures to stop COVID-19 spread (e.g., through closing business and halting travel), it is important to note that the actual death totals from COVID-19 were outside the IHME model’s 95% confidence interval 70% of the time [15]. This fact is notable given the importance of deaths as a measurement of COVID-19 spread during the early days of the pandemic [16]. Yet, predictive validity is an ex post consideration for model selection. Policymakers must make best possible decisions without knowing the future or the predictive validity of any given model. Thus, in the historical case study, the selection of models best used for a given jurisdiction was based on the factors noted in Section 2.

Allocation, logistics, and utilization of personal protective equipment (PPE) during the initial response to COVID-19 was another use of COVID-19 models by policymakers. Hospital administrators and policymakers need to accurately account for burn rates of PPE (e.g., masks, surgical gowns, and facemasks) to request appropriate funding from their funding sources. The technical team used the University of Basel model for informing PPE. Regarding stockpiling respirators, estimations of need were essential to decreasing over-stocking which may diminish the supply in other areas of need, or under-stocking respirators, which would have had severe consequences.

#### 3.1.2. Use Case 2: Isolation and Quarantine Capacity Planning

In the second surge that Hawaii experienced in the fall of 2020, the models adapted and used through the HiPAM work group were used to communicate the forecasted number of cases, hospitalizations, and deaths, primarily through behind-the-scenes communications to senior state policymakers including the Governor’s office and the county Mayors, among others. Whereas the models used for hospital bed planning were informing HI-EMA as a key state agency, the departure of the epidemiologic advisor to HI-EMA in July of 2020 resulted in the HiPAM work group stepping in to serve as the go-to local institutional contact for modeling, supported by the Hawaii Data Collaborative and Hawaii Department of Health Behavioral Health Administration. HiPAM had formed in April of 2020, bringing together health professionals, data scientists, mathematicians, and agency staff to convene around an agenda on COVID-19 modeling. Given the limitations in resources in a small remote state, there was a need to pool resources and efforts together to reduce duplication and confusion. The interdisciplinary HiPAM work group was structured on past work of the HiPAM chair, who had previous experience using work groups at a think tank in Washington, DC (the Center for Global Development). Based on the work of the HI-EMA epidemiologic advisor and technical team with support from HiPAM, a need for an ongoing forecast for the state was identified. By July of 2020, HiPAM launched an online two-week COVID-19 forecast, accessible publicly.

As the local response evolved including increasing capacity for testing, tracing, and isolation and quarantine, the models were also used to inform isolation and quarantine capacity which had a particular emphasis on vulnerable populations including homeless individuals, Native Hawaiian and Other Pacific Island communities, as well as individuals with co-occurring mental illness and substance use challenges. The Hawaii Department of Health’s Behavioral Health Administration (BHA) was designated to lead isolation and quarantine beginning in August 2020 as Hawaii experienced its second surge. The BHA was also the sole DOH unit to establish the standalone Temporary Quarantine & Isolation Center specifically for homeless individuals and later for medically needy individuals [17]. As the BHA leadership actively participated in HiPAM, BHA leadership had sought inputs and guidance from HiPAM models to monitor adequacy of bed capacity for isolation and quarantine real-time case counts. Models from HiPAM were also used to estimate the adequacy of shelter capacity for homeless populations.

There was a policy need for a simple benchmark to identify whether there was adequate isolation and quarantine capacity and, specifically, enough beds procured by the State of Hawaii. With limited time and resources available to conduct a detailed epidemiologic and demographic study, there was a need to identify in a simple manner how many people might need isolation and quarantine services. Eligibility for isolation and quarantine services in a government-procured hotel was determined in part based on whether an individual was able to safely isolate at home and whether the individual lived in a shared bedroom with someone. In Hawaii, the percentage of the population living in a shared bedroom was identified to be nearly 10%. When applied to the total number of active COVID-19 cases at any given time, this benchmark helped to inform the planning for the total beds procured by the State of Hawaii for isolation and quarantine operational activities in 2020. Although ‘active cases’ as a concept was challenging to independently measure due to lack of capacity for verification of individuals released from isolation and quarantine, the rolled-up cumulative case count over the last 14 days was used as a proxy for active case count for the state, on which the ten percent was applied to estimate need for isolation and quarantine outside of one’s home.

#### 3.1.3. Use Case 3: Broad Public and Media Communications

In the Delta and Omicron surges in 2021 in the summer and winter, respectively, HiPAM took a direct public communications strategy to communicate the results of the model and forecast. Rather than use only backdoor communication with senior policymakers and government authorities, HiPAM emphasized direct communications with the media, similar to the weatherman, as well as the release of regular and timely reports sent to all key policymakers and news outlets in the state, supplementing the online web tool hosting the two-week advance forecast, which had been launched in July of 2020.

By 2021, a locally developed and customized model led by mathematicians (Chyba et al.) became the de facto and well-accepted model by HiPAM for the state, and the other models by the University of Basel and IHME were abandoned by 2021 [18]. The Chyba et al. model fulfilled the key considerations for the models including local parameter customizability, local age distribution, use for assessing different policy scenarios, and being customizable and potentially open source because it was developed in-house. Developing local in-house mathematical and epidemiologic modeling capacity is extremely challenging and dependent upon the availability of scientific experts willing to engage in real-world policy challenges and was spearheaded by funding from the National Science Foundation competitively awarded to Chyba et al. [18].

The direct public dissemination of the model results to the state in 2021 and 2022 was vastly different from the 2020 approach of behind-the-scenes information provided to senior leaders. Nevertheless, this public communication strategy also had challenges and risks in terms of the ways in which the modeling results and information was communicated and the kinds of questions and concerns posed by the media, policymakers, and the public. The media and policymakers, for example, repeatedly asked HiPAM representatives challenging questions about the specific policy guidance that should be made based on the modeling results. Yet in order to ensure and maintain the scientific credibility of the HiPAM models, HiPAM repeatedly emphasized its role as a scientific body that focused on high-quality models based on best-available evidence and ever-evolving science. It had to remind the media, the public, and policymakers that while HiPAM’s information was important, policymakers would need to use multiple sources of information, in order to make decisions. In doing so, HiPAM reinforced its role as a scientific body and not as a body making policy recommendations or actions. Maintaining a scientifically neutral and unbiased position was essential for ensuring HiPAM public credibility and recognition.

A second key question repeatedly raised by the public, the media, and policymakers was about the time horizon of the model. There was a longstanding desire for understanding the forecast or projection of COVID-19 well into the future by several months. HiPAM, however, maintained a stance of emphasizing a two-week forecast horizon, and that anything longer than that would be subject to change. Seeing the real-world mistakes of models communicated at the national level, HiPAM made a deliberate choice to focus on a limited time horizon and repeatedly emphasized the ways in which individual and policy actions could easily influence the forecast beyond two weeks.

A third key challenge of direct communications was the emphasis on the dynamic nature of the modeling results. HiPAM repeatedly noted that upon release of the forecast, the projection would immediately change based on the fact that knowledge and information about the situation would result in changes in individual behavior as well as policy changes and action. Unlike weather forecasting, dissemination of a COVID-19 forecast changes the forecast itself. This difficulty in communicating the dynamic nature of modeling was challenging throughout the pandemic, even with seasoned media reporters and engaged policymakers and legislators.

The media and communications engagements was also broad through numerous media engagements through print, television, radio, social media, state and county government hearings and fora, and so on, raising public awareness of the COVID-19 modeling writ-large beyond the closed circles of policymakers. The media strategy led to wide acceptance, recognition and use of the COVID-19 modeling not only by state agencies but also other health care provider organizations including the local hospital association. The credibility and validity of the HiPAM model work was emphasized primarily through maintaining a neutral stance on any specific policy recommendation but focusing on the specific technical result or information that the model provided.

The interactive communication loop also enabled immediate feedback from the policymakers, the media, and the public to ask questions including about understanding the potential impacts of a particular policy or intervention scenario. These communication engagements also enabled policymakers to request and clarify potential scenarioing requests for any given policy.

### 3.2. Using Models for Reopening Amidst Decline

Models were also used to inform policy decisions for reopening, and these decisions were equally pressing due to the economic impacts of COVID-19. In the case of Hawaii, one major question facing policymakers seeking to reopen is how and when travel volumes, both domestic and international, can increase. Some of the early mechanistic models only accounted for a population where the total size stayed the same as well as for how COVID-19 would progress under certain mitigation efforts scenarios pre-programmed into the model. Moreover, there continues to be uncertainty and evolving understanding about the basic scientific facts and assumptions of COVID-19 (e.g., extent of screening for asymptomatic transmission [19] and the infection fatality rate [20]), making policy decisions difficult. As travel volumes return to higher levels, models that factor in imports of new cases can provide more accurate estimates of travel impacts on overall disease spread. Teams engaged in epidemic forecasting can estimate metrics for different travel volume scenarios and demonstrate how the range of new cases is dependent on how many imported cases are brought into their community.

There are many assumptions built into the various COVID-19 models, such as whether symptomatic travelers will restrict themselves from traveling and whether they will be identified at the port of departure. Arguably, one of the largest considerations for developing travel scenarios is that of asymptomatic and pre-symptomatic cases—how assumptions about these parameters are incorporated into a given model, the distribution of COVID-19 cases which are asymptomatic or pre-symptomatic, and the rate of spread from these cases [21,22,23]. Reopening strategies based on one or multiple tests have been suggested without any numerical estimations of possible infected travelers slipping through. Modeling can provide policymakers with an educated guess when comparing reopening strategies based on frequency and type of tests.

Testing, contact tracing, and isolation and quarantine represent major public health tools for policymakers responding to COVID-19. States can consider how tests such as body temperature and symptom screens as well as standard polymerase chain reaction (PCR) tests can be linked to travel policies, and use models to help estimate the potential impacts and consequences of different testing strategies.

Most models at present do not account for health impacts beyond the immediate COVID-19 health impacts such as those pertaining to mental health, reductions in use of other essential health services, or long-term care facilities or other congregate settings. Mental health and substance use, already important public health issues prior to COVID-19, have become exacerbated by secondary and tertiary impacts due to COVID-19 [24].

COVID-19 will continue to directly impact communities and indirectly for decades to come. Policymakers will need to shift from the use of models that focus on hospital capacity and reopening, to models identifying long-term health and economic impacts of COVID-19, such as mental health, access to non-COVID-19 health care services, education, and other dimensions of the social determinants of health. Most of the COVID-19 models used at present have not directly incorporated these long-term impacts. The use of Bayesian modeling and synthetic population models can and have begun to be used to examine these longer-term health impacts and policy implications, as this type of modeling accounts for additional differences in a population such as economic status and race.

## 4. Discussion

Epidemiologic models used for COVID-19 are numerous and complex, requiring subject matter experts to appropriately utilize data, interpret, and communicate results. The study used a case study approach to provide a historical account of the events and reflect on the lessons learned from one jurisdiction in the United States. The historical events reflected in this case study demonstrate the real-world challenges that policymakers and subject matter experts face when deciding which model to use, including demonstrating how even with accurate data, utilization of an inappropriate model or considerations has the potential to lead to inappropriate interpretation of results. COVID-19 models vary by their designed intent and understanding these differences, including their differences in geographic application and applicability to specific policy decisions, is necessary for policymakers to better utilize them in making decisions [25,26,27].

There are several key lessons that can be drawn from this case study documenting the historical application of mathematical and epidemiologic models for key policy decisions. First, COVID-19 modeling in Hawaii benefited from the incorporation of state-specific data which were historically argued to directly result in cost savings from decreased unnecessary spending, particularly in the case of the hospital capacity planning in the early part of 2020 during the pandemic. This model incorporated two of the most important factors that assist local leaders in modeling local issues, age distribution and customization that was specific to Hawaii [9,10]. It also helped to inform isolation and quarantine planning and adequacy of facilities available in order to meet demand and need in the fall of 2020, as well as helped to inform the media, the public, and policymakers of the potential magnitude of the Delta and Omicron surges in 2021 to 2022.

Second, regardless of model selection, it is essential that model outputs be interpreted directionally, not as a forecast of hard, immutable numbers, and with a clearly delineated time horizon. The numerous known and unknown factors and their combinations thereof impacting the spread of COVID-19 means that no model, no matter the level of sophistication, can concurrently accurately account for all factors. Further, the nature of the models, easily influenced by actions of individuals and policies today, makes the models dynamic and uncertain rather than static, despite a desire for static and definitive answers. Therefore, numbers produced regarding cases, hospitalizations, deaths, etc. should be communicated and understood as a possible scenario should current trends continue forward into the future assuming no change in policy or human behavior, which is an impossible assumption to begin with as soon as a forecast is released and disseminated.

Third, when the above factors are properly considered and both model outputs—projected trends and subsequent reductions by means of interventions—are combined, it is essential that warnings are heeded and action be taken as soon as possible. Based on appropriate interpretation of a model, policymakers can be advised if a policy intervention may avert critical thresholds such as hospital and ICU capacity. With these crucial timings in mind, policymakers can then use other models to help more accurately understand how an intervention may impact the Rt in a given population and, subsequently, what scenarios of intervention combinations and efficacies might result in faster control or even elimination of COVID-19 [28]. Hesitation in implementation or inadequate interventions can have dramatic effects on disease spread, such as the delayed and scattered approach to mask wearing early on in the pandemic [29]. However, failure to grasp model limitations can also result in hasty and expensive overreactions.

Fourth, these models require a firm grasp of epidemiologic concepts. As such, policymakers are advised to seek out and involve an interdisciplinary scientific advisory group as early as possible to translate modeled outcomes into actionable context. Because of the complexity of models, significant unpredictable impact of human behavior, and the potential for misinterpretation, it can be argued that these models do more harm than good. Rather than dismiss the use of models because of their complexity, policymakers should incorporate into the scientific and technical advisory group or ‘brain trust’ as early as possible to help inform and navigate the difficult policy decisions that can have positive impacts on their constituents and communities.

In the case study herein, the brain trust was a diverse team that provided input from various areas of expertise (e.g., epidemiology, data science, behavioral health, and mathematics). The role of the local university to work in close collaboration and partnership with the community and government authorities is essential. While it remains to be seen what the long-run impact of the communication of COVID-19 was modeling and forecasting in the state of Hawaii, it cannot be denied that there is value of having mathematical and epidemiologic capacity of scientific experts who are willing and able to communicate and contribute to real-world policy challenges in service of the public and community during an extended period of confusion and crisis.

The work in Hawaii of using a brain trust may be contextualized to the work in many countries around the world which used COVID-19 modeling and forecasting to inform decision making. In Hawaii, the creation of HiPAM included a range of local experts from epidemiology, public health, data science, and mathematics who were able to contribute to modeling and forecasting locally. Other countries such as Ireland, United Kingdom, New Zealand, and several others had technical advisory groups that provided inputs and information to policymakers who ultimately made the policy calls and decisions. For example, in New Zealand, a COVID-19 technical advisory group comprised medical, public health, and academic advisors, which provided advice to the ministry of health. In Australia, the COVID-19 Expert Database hosted by the Australian Academy of Science provided a mechanism for governments and decision makers to have easier access to expertise in COVID-19. The UK also established a Scientific Advisory Group for Emergencies as the entity responsible for providing scientific advice to UK decision makers while not representing official government policy.

More research is needed to examine how to create and then institutionalize these bodies with blended technical expertise, savvy communication skills, and linkages to policymakers making decisions. What is the optimal composition of these bodies? To what extent are these bodies linked and connected to decision-making? What is the role and balance of neutrality in balancing scientific facts relative to policy recommendations? The work in Hawaii serves as a basis for the hypothesis that the composition of a body that draws from a wide range of expertise beyond clinical medicine or public health fields can help to bridge challenges of mathematics, applied or real-world epidemiology, behavioral health, and data science. Existing public health institutions such as those pertaining to health technology assessment or epidemic intelligence are also relevant and should be considered before new bodies are duplicated, creating more institutional fragmentation, siloization, and duplication of effort.

Further, there is a need for communication and practitioners who can help to translate and communicate complex ideas into simple concepts for policymakers and the public. We would also hypothesize that whether forecasting and modeling are sidelined or are integral to decision-making depends on leadership and governance in formally supporting, building, seeking guidance from, and incorporating information from technical advisory groups. During a time when there is controversy and doubt about science, the ways in which scientific and technical advisory bodies monitor ever-evolving science and evidence and how such bodies intersect with and communicate with policymakers who then make decisions for policies, programs, and practice merits further study.

There is some research in the field of political science and public administration which examines the ways in which policy decisions and actions are determined and implemented. Political science theories have been applied to understand how political actors make policy actions and political decisions, including Kingdon’s Multiple Streams Model [30] and Reich’s work on political economy [31]. Work by Walt et al. noted that rigorous health policy research methods have much to be desired for understanding the policy process [32]. In particular, a major limitation of this historical case study is its focus on a single jurisdiction—the state of Hawaii—one that lacks a comparison or historical “counterfactual” for what might have happened in the absence of this work on modeling in the state. We argue that drawing on historical perspectives and chronology from lived experiences of those engaged in real-world implementation and operations (albeit informed by modeling and evidence) is a research methodology.

This case study also did not explicitly examine cases of misappropriation and misuse of models in Hawaii or cases in which the modeling outputs were ignored or otherwise not used for specific policy actions. Determining what constitutes misuse and misappropriation is beyond the scope of this paper, but we acknowledge that the complexity of models makes inappropriate or poor application quite possible, if not the default. Future research would be valuable to examine the different ways in which modeling informed or did not inform key policy decisions in multiple states and jurisdictions and the variations in communication about modeling.

## 5. Conclusions

This article has emphasized the role of localizing knowledge that can be translated and used to inform local decisions. With tremendous uncertainty about a novel disease, the need for thoughtful application of scientific knowledge is ever more pressing. Although the specific use cases and policy window and moment for critical decisions described herein have now passed, the lessons from this case study may be relevant for jurisdictions seeking to make smarter decisions informed by modeling. The knowledge and experience that was gained through these lived experiences may be applicable to island countries and states with age, ethnicity, and other sociodemographic distributions similar to Hawaii. The knowledge and experience from this case study may also help to inform jurisdictions experiencing limitations in resources, time, and scientific expertise for COVID-19 modeling in informing policymaking.

## Figures and Tables

**Table 1 ijerph-19-06119-t001:** Landscape of selected models for informing COVID-19 control and mitigation, 2020.

	Objective of Model	Localized Customizability	Local Age Distribution	Type of Model	Open Source
IHME [5]	Estimate hospital impacts	No	Unknown ^1^	Statistical	No
Imperial College London [6]	Assess public health measures on spread	No ^2^	No ^2^	Mechanistic	No ^2^
Epidemic Calculator [7]	Estimate change in epi curve after reduction in transmission	Yes	No	Mechanistic	Yes
University of Basel [8]	Planning tool with features such as imported cases and age groups	Yes	Yes	Mechanistic	Yes

^1^ The IHME model was closed source so it was unknown how local age distribution was taken into account. ^2^ The source code was not available when the original Report 9 was released. The updated source code was eventually made available much later with limited documentation, making localized use of the model difficult.

**Table 2 ijerph-19-06119-t002:** Selected model assumptions for informing COVID-19 control and mitigation, 2020.

	Key Assumption #1: Asymptomatic vs. Symptomatic	Underestimate or Overestimate on Total Severity (Cases, Deaths)	Key Assumption #2: Age Distribution	Underestimate or Overestimate on Total Severity (Cases, Deaths)	Other Assumptions
IHME [5]	As the model is not open source, it is unapparent to what extent asymptomatic vs. symptomatic is taken into account	As the model is not open source, it is unapparent to what extent asymptomatic vs. symptomatic is considered	Uses actual data and are based on results for specific age distributions (for China and Italy) applied and adapted to other populations	As the model is not open source, it is unapparent how the age-specific distributions are incorporated and applied	
Imperial College [6]	Does not appear to distinguish between asymptomatic and non-hospitalized symptomatic individuals	Same as for Epidemic Calculator (see below)	Agent based model has individuals that reflect the population’s age distribution	Not applicable	Assumes changes in transmission are reflected through mobility of the population
Epidemic Calculator [7]	Does not appear to distinguish between asymptomatic and non-hospitalized symptomatic individuals	May underestimate total severity as asymptomatic individuals are more likely to spread COVID-19 as they are unaware, they are infected and/or infectious	Does not take age or age distributions into account and unclear the reference population or data used to benchmark (e.g., China)	May overestimate hospitalizations and fatalities if population is younger, as increased age significantly increases risk ^1^	
University of Basel [8]	Does not appear to distinguish between asymptomatic and non-hospitalized symptomatic individuals	Same as for Epidemic Calculator (see above)	Divides population into age groups with age-group-specific parameters (such as how severe, critical, and fatal the infection is)	Depends on whether the user correctly selects the age distribution and age-group-specific parameters of geographic location of interest	Puts imported cases into the Exposed compartment, which can be interpreted as the cases coming from outside are all incubating/recently infected and not symptomatic

^1^ The United States has a younger age distribution compared to China, so models that use aggregate estimates of mortality for China may overestimate mortality for the United States unless age-specific mortality distributions are accounted for.

## Data Availability

Not applicable.

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
