# Peer review of "Mitigation Planning and Policies Informed by COVID-19 Modeling: A Framework and Case Study of the State of Hawaii"

_ijerph, 2022, doi:10.3390/ijerph19106119_

Round 1
Reviewer 1 Report
Mitigation Planning and Policies Informed By COVID-19 Modeling: A Framework and Case Study of the State of Hawaii
This paper is very interesting for the scientific community as well as for the public in general. It analyses four COVID-19 models used by the state of Hawaii public health authorities to guide in implementing mitigation measures and/or reopening measures in 2020. The four models comprise the University of Washington Institute for Health Metrics and Evaluation (IHME) model; the Imperial College London model, the Epidemic Calculator, and the University of Basel model. In Hawaii, the Hawaii Emergency Management Agency (HI-EMA) sought help from experts in epidemiology to analyze the results of the four models. One policy implication from epidemiologic COVID-19 modeling in Hawaii is the incorporation of local detailed data resulted in cost savings from using only the necessary measures.
Comments and suggestions:
- The abstract ha to be rewritten in order to make it clear the methodology and results of the manuscript. It is very vague as it is.
- The last two paragraphs of the 1. Introduction have to be rewritten in order to describe the methodology and the data. The case study is not well explained, we could not understand what you were explaining.
- You have to write a paragraph in the end of the 1. Introduction describing the structure of the manuscript.
- Section 2. Can be included in the introduction. Please remove the numbering.
- Section 3 may be renamed “COVID-19 Models used by Policymakers”
- In the second paragraph of section 3. You write: “We present the case of Hawaii to illustrate the application of the use of 90 models for specific policy decisions.” However, we could not understand the case, nor its application. Please show/summarize the data used and what you did to obtain the results.
- You need to make a brief description of table2. There are also some inconsistencies when comparing Table 1 with 2. For example, in table 1, you mention in the column of “Local Age Distribution” that is unknown or No. In table 2, you describe in detail the assumptions. Please clarify these inconsistencies.
- In section 4. Interpreting Results, we have to ask where are the results?, what did you do? Please show/summarize the results (see point 6). These two paragraphs should be included in the discussion.
- The section 5. Application of Models to State Policy Decisions, is the section usually referred as the “results”, since you are applying the data of Hawaii to the IHME model. Here you have to summarize the results of your data in a table, for us to understand how you arrived to the policies.
Author Response
Reviewer #1
Mitigation Planning and Policies Informed By COVID-19 Modeling: A Framework and Case Study of the State of Hawaii
This paper is very interesting for the scientific community as well as for the public in general. It analyses four COVID-19 models used by the state of Hawaii public health authorities to guide in implementing mitigation measures and/or reopening measures in 2020. The four models comprise the University of Washington Institute for Health Metrics and Evaluation (IHME) model; the Imperial College London model, the Epidemic Calculator, and the University of Basel model. In Hawaii, the Hawaii Emergency Management Agency (HI-EMA) sought help from experts in epidemiology to analyze the results of the four models. One policy implication from epidemiologic COVID-19 modeling in Hawaii is the incorporation of local detailed data resulted in cost savings from using only the necessary measures.
Comments and suggestions:
- The abstract ha to be rewritten in order to make it clear the methodology and results of the manuscript. It is very vague as it is.
Authors’ response: Thank you for this helpful feedback. We have substantially revised the abstract to make the policy methodology and policy results explicit since it is a policy case study, which is different from an epidemiological study with standard methods and data. The focus on the methodology and results is the specific policy decisions and lived experiences that the authors had in informing and shaping policy decisions. Please see the abstract as follows:
Abstract: In the face of great uncertainty and a global crisis from COVID-19, mathematical and epidemiologic COVID-19 models proliferated during the pandemic. Yet many models were not created with the explicit audience of policymakers, the intention of informing specific scenarios, or explicit communication of assumptions, limitations, and complexities. This study presents a case study of the roles, uses, and approaches to COVID-19 modeling and forecasting in one state jurisdiction in the United States. Using a review of the historical real-world events through lived experiences, we first examine the specific modeling considerations used to inform policy decisions. Then we review the real-world policy use cases and key decisions that were informed by modeling during the pandemic including the role of modeling in informing planning for hospital capacity, isolation and quarantine facilities, and broad public communication. Key lessons are examined through the real-world application of modeling include the importance of locally tailored models, the role of a ‘brain trust’ or technical advisory group, and the challenges of communicating technical considerations to a broad public audience.
- The last two paragraphs of the 1. Introduction have to be rewritten in order to describe the methodology and the data. The case study is not well explained, we could not understand what you were explaining.
Authors’ response: Thank you. We have revised the last two paragraphs to explicitly account for the policy methodology and data with a focus on the historical timeline of events used for this policy case study as follows:
This study presents a real-world historical case study of the roles, uses, and approaches to COVID-19 modeling and forecasting for policy decisions and policy use cases, drawing from the historical record in one state jurisdiction in the United States. This study does not present a micro-level analysis of detailed modeling and its mathematical specifications, but rather provides a macro-level historical and policy perspective on the ways in which modeling informs policymaking. The methodology and data used for this case study rely on a review of the historical facts and real-world events through lived experiences of the authors of this paper who are members of the Hawaii Pandemic Applied Modeling Work Group (HiPAM) (https://www.hipam.org), the Hawaii Emergency Management Agency (HI-EMA), or the Hawaii Department of Health in a variety of capacities during the pandemic from March 2020 to May 2022.
This case study is intended to and may help future policymakers seeking to navigate this complex landscape of models and draw upon practical lessons learned on how to make appropriate evidence-based decisions using models. As such, this paper is structured as follows. The first part of this case study focuses on the major technical considerations of mathematical and epidemiologic models and how models were selected given real-world limitations of time and resources. The second part of this case study reviews and summarizes the real-world policy use cases and key policy decisions informed by modeling during the pandemic surge and decline, including the role of modeling in informing planning for hospital capacity, isolation and quarantine facilities, and broad public communication. on the lived experiences of authors. We then reflect on the key lessons and discussion from the use cases that may be relevant for other jurisdictions seeking to use modeling to inform decision-making.
- You have to write a paragraph in the end of the 1. Introduction describing the structure of the manuscript.
Authors’ response: Thank you for this valuable suggestion. We have revised the last paragraph of the Introduction describing the structure of the manuscript as follows:
This case study is intended to and may help future policymakers seeking to navigate this complex landscape of models and draw upon practical lessons learned on how to make appropriate evidence-based decisions using models. As such, this paper is structured as follows. The first part of this case study focuses on the major technical considerations of mathematical and epidemiologic models and how models were selected given real-world limitations of time and resources. The second part of this case study reviews and summarizes the real-world policy use cases and key policy decisions informed by modeling during the pandemic surge and decline, including the role of modeling in informing planning for hospital capacity, isolation and quarantine facilities, and broad public communication. on the lived experiences of authors. We then reflect on the key lessons and discussion from the use cases that may be relevant for other jurisdictions seeking to use modeling to inform decision-making.
- Section 2. Can be included in the introduction. Please remove the numbering.
Authors’ response: Thank you for this suggestion. We have incorporated this into the introduction.
- Section 3 may be renamed “COVID-19 Models used by Policymakers”
Authors’ response: Thank you, we have renamed it and also renumbered it to Section 2.
- In the second paragraph of section 3. You write: “We present the case of Hawaii to illustrate the application of the use of 90 models for specific policy decisions.” However, we could not understand the case, nor its application. Please show/summarize the data used and what you did to obtain the results.
Authors’ response: Thank you for this valuable feedback. We have clarified the specific case study of the historical policy use case and scenario in which the models were selected and used and reflect the data sources being the model information and documentation made available publicly. The revised text is as follows:
2.1. Model selection
A “model” refers to a mathematical or logical representation of the biology and epidemiology of disease transmission and its associated processes [4]. To date there are more than 40 COVID-19 models available. In many locations around the world, there was a need for timely public health decisions pressing against the lack of available time, resources, and expertise – and in the case of Hawaii and likely in other locations as well, a severe dearth of epidemiologists in the jurisdiction. Thus, it was not feasible nor practical for policymakers and their technical support teams to comprehensively review all models in order to make decisions. Instead, policymakers were continuously forced to make strategic decisions to select and use tools in order to make the best available information at hand. Nevertheless, even the selection of tools in order to make policy decisions requires technical expertise in order to wade through the science and complexity.
The first part of this case study focuses on the major criteria for selecting a model. Beginning in April of 2020, the HI-EMA tasked a technical team including a lead epidemiologic adviser, a technical analyst, and a working group serving as a technical advisory body to review and use models that would be used to inform a variety of specific policy decisions, described in the second part of this case study. The four models that were ultimately chosen and used for informing Hawaii public authorities for decision making were based on selective review rather than comprehensive review of models. The four models used were the following: the University of Washington Institute for Health Metrics and Evaluation (IHME) model [5], the Imperial College London model [6], the Epidemic Calculator [7], and the University of Basel model [8]. The dimensions for reviewing and selecting these models are described in section 2.2.
Upon review of the documentation and source code, if available, of these models, in 2020, the technical team identified some of the key assumptions of these models. These assumptions were crucial in understanding the limitations or applicability of a given model to a particular jurisdiction, and in this case, the state of Hawaii. These assumptions and limitations are discussed in section 2.3.
- You need to make a brief description of table2. There are also some inconsistencies when comparing Table 1 with 2. For example, in table 1, you mention in the column of “Local Age Distribution” that is unknown or No. In table 2, you describe in detail the assumptions. Please clarify these inconsistencies.
Authors’ response: Thank you for identifying these issues, and we have provided a brief description of Table 2 as follows:
2.3. Model assumptions and limitations
This section reviews the key assumptions identified by the technical team of these models, which were used to inform the applicability of a given model to a particular jurisdiction, and in this case, the state of Hawaii. Table 2 presents explicitly some of the assumptions in the four models. Ultimately, the technical team chose to use the Epidemic Calculator and University of Basel model for several key decisions in 2020, described in section 3, and then later relied on a locally customized and developed model beginning in 2021.
Regarding the inconsistencies noted by the reviewer, we have provided clarification for the inconsistencies measured, the Local Age Distribution column in Table 1 refers to whether or not the model accounts for the specific age distribution of a local area or state jurisdiction (such as the state of Hawaii). This was of specific interest for our technical team and policymakers due to the different local age distribution in Hawaii, and would be something of interest for other states and areas as well. Table 2 details if and how the models accounted for age distribution in any form, and not whether they specifically allow or account for the local age distribution of Hawaii or another local area.
- In section 4. Interpreting Results, we have to ask where are the results?, what did you do? Please show/summarize the results (see point 6). These two paragraphs should be included in the discussion.
Authors’ response: Thank you for your comment, which has helped us to greatly rework and revise our paper structure. The comment has also helped us to understand and clarify that the overarching purpose of the case study is not to provide a detailed micro-level study of the models, but rather to review the historical chronology and timeline for how the models were used for macro-policy decisionmaking. We have reorganized the structure of case study to first focus on the key model considerations in section 2, with section 3 focused on the applications of the models to specific policy use cases. In addition, we have incorporated these paragraphs as the starting paragraph of the Discussion section as follows.
Epidemiologic models used for COVID-19 are numerous and complex, requiring content experts to appropriately utilize data and interpret results. The study used a case study approach to reflect on the lessons learned from one jurisdiction in the United States. The historical events reflected in this case study demonstrates the real-world challenges that policymakers face when deciding which model to use, including demonstrating how even with accurate data, utilization of an inappropriate model and/or considerations may lead to inappropriate interpretation of results. COVID-19 models vary by their designed intent and understanding these differences, including their differences in geographic application and their application to specific policy decisions, is necessary for policymakers to better utilize them in making decisions [25–27].
- The section 5. Application of Models to State Policy Decisions, is the section usually referred as the “results”, since you are applying the data of Hawaii to the IHME model. Here you have to summarize the results of your data in a table, for us to understand how you arrived to the policies.
Authors’ response: Thank you so much for your comment and feedback. As noted above, we have focused this section to be on the results of the specific policy use cases. As this is a policy case study, we delved into the specific policy use cases and policy considerations that were used to inform the policy decisions and the ways in which epidemiologic information and technical guidance was used to inform policies.
- Policy Use Cases of Applying Models to Specific State Policy Decisions
In this section of the policy case study, we document the historical record for how models were used to inform key policy decisions in the jurisdiction, both in terms of managing capacity during a surge and reopening amidst a decline. The policy case study reflects actual lived experiences along with (now-historical) observations and perspectives of the technical team and the working group who were providing information to policymakers seeking to make decisions informed by modeling. Thus, the detailed micro-level analyses for each use case are not presented herein, but rather the specific translation of evidence to knowledge and communication with a variety of stakeholders including policymakers, media, and the public are described.
Reviewer 2 Report
Authors conducted a case study on COVID-19 forecasting models practically well and timely. A number of practioners can achieve lessons on pros and cons of four models that authors presented in this case study. If authors can improve followings issues appropriately, the quality of this practical research could be published in IJERPH well.
First, the process of selecting the four models should be elaborated more clearly. Also, authors should provide the explanation on research methods more deeply and must follow the social scientific approach. Current manuscript is not sufficiently explained.
Second, authors need to provide more clear contrast and comparison of their findings to other cases of States as well as other countries. Current report is not that much discussing and presenting in-depth discussions. Authors need to write up more implications on their research.
Lastly, as an academic research paper, authors need to look up the topic with appropriate theoretical viewpoints. In other words, they need to conduct more academic literature reviews and provide academical implications to the readers who want to understand the meaning of this research in terms of academical research.
In sum, the topic of this research is meaningful and timely well conducted so that I believe that authors can improve this manuscript well. Thank you.
Author Response
Reviewer #2
Comments and Suggestions for Authors
Authors conducted a case study on COVID-19 forecasting models practically well and timely. A number of practioners can achieve lessons on pros and cons of four models that authors presented in this case study. If authors can improve followings issues appropriately, the quality of this practical research could be published in IJERPH well.
Authors’ response: Thank you for your feedback, and we agree that this research is very practical. We see that there is some confusion about the scope of the study, which is not intended to focus on the epidemiologic considerations only of the four models, but rather to focus on the specific policy use cases of these models. We realize that the methodology of the study is a policy research methodology focused on historical review and lived experiences, which is not a typical or standard epidemiologic study. As such, we have expanded the section describing the policy research methodology as follows in the introduction:
This study presents a real-world historical case study of the roles, uses, and approaches to COVID-19 modeling and forecasting for policy decisions and policy use cases, drawing from the historical record in one state jurisdiction in the United States. This study does not present a micro-level analysis of detailed modeling and its mathematical specifications, but rather provides a macro-level historical and policy perspective on the ways in which modeling informs policymaking. The methodology and data used for this case study rely on a review of the historical facts and real-world events through lived experiences of the authors of this paper who are members of the Hawaii Pandemic Applied Modeling Work Group (HiPAM) (https://www.hipam.org), the Hawaii Emergency Management Agency (HI-EMA), or the Hawaii Department of Health in a variety of capacities during the pandemic from March 2020 to May 2022.
This case study is intended to and may help future policymakers seeking to navigate this complex landscape of models and draw upon practical lessons learned on how to make appropriate evidence-based decisions using models. As such, this paper is structured as follows. The first part of this case study focuses on the major technical considerations of mathematical and epidemiologic models and how models were selected given real-world limitations of time and resources. The second part of this case study reviews and summarizes the real-world policy use cases and key policy decisions informed by modeling during the pandemic surge and decline, including the role of modeling in informing planning for hospital capacity, isolation and quarantine facilities, and broad public communication. on the lived experiences of authors. We then reflect on the key lessons and discussion from the use cases that may be relevant for other jurisdictions seeking to use modeling to inform decision-making.
First, the process of selecting the four models should be elaborated more clearly. Also, authors should provide the explanation on research methods more deeply and must follow the social scientific approach. Current manuscript is not sufficiently explained.
Authors’ response: Thank you so much for your feedback. We realized from your comments and the feedback of Reviewer #1 that there was an expectation that this article might be structured as a standard epidemiological article. Instead, because this is a policy case study drawn from lived experiences and historical chronology in the policy space, we use the methodology and data of reviewing the historical circumstances and constraints of decisions. We have provided more explanation on the explicit role of lived experiences in providing the methodology and data for the policy circumstances as follows. The methodology for the model selection is now explicitly documented in section 2 as follows:
The first part of this case study focuses on the major criteria for selecting a model. Beginning in April of 2020, the HI-EMA tasked a technical team including a lead epidemiologic adviser, a technical analyst, and a working group serving as a technical advisory body to review and use models that would be used to inform a variety of specific policy decisions, described in the second part of this case study. The four models that were ultimately chosen and used for informing Hawaii public authorities for decision making were based on selective review rather than comprehensive review of models. The four models used were the following: the University of Washington Institute for Health Metrics and Evaluation (IHME) model [5], the Imperial College London model [6], the Epidemic Calculator [7], and the University of Basel model [8]. The dimensions for reviewing and selecting these models are described in section 2.2.
Upon review of the documentation and source code, if available, of these models, in 2020, the technical team identified some of the key assumptions of these models. These assumptions were crucial in understanding the limitations or applicability of a given model to a particular jurisdiction, and in this case, the state of Hawaii. These assumptions and limitations are discussed in section 2.3.
Second, authors need to provide more clear contrast and comparison of their findings to other cases of States as well as other countries. Current report is not that much discussing and presenting in-depth discussions. Authors need to write up more implications on their research.
Authors’ response: Thank you for this feedback. The scope of this case study is specific to a particular jurisdiction (the state of Hawaii) and multiple policy decisions that were informed by modeling. As a result, we have substantially elaborated on the two other use cases in section 3 as follows:
Use Case 2: Isolation and quarantine capacity planning
In the second surge that Hawaii experienced in the fall of 2020, the models adapted and used through HiPAM were used to communicate the forecasted number of cases, hospitalizations, and deaths, primarily through behind-the-scenes communications to senior state policymakers including the Governor’s office and the county Mayors, among others. Whereas the models used for hospital bed planning were informing HI-EMA as a key state agency, the departure and changing role of one technical advisor from a role as epidemiologic advisor to HI-EMA in July of 2020 resulted in the HiPAM filling to serve as local institutional knowledge for modeling, supported by the Hawaii Data Collaborative and Behavioral Health Administration. HiPAM had formed in April of 2020, bringing together health professionals, data scientists, mathematicians, and agency staff to convene around an agenda on COVID-19 modeling. Given the limitations in resources in a small remote state, there was a need to pool resources and efforts together to reduce duplication and confusion. Based on the work of the HI-EMA epidemiologic advisor, the need for an ongoing forecast for the state evolved, and by July of 2020, HiPAM launched an online two-week COVID-19 accessible publicly.
As the local response evolved including increasing capacity for testing, tracing, and isolation/quarantine, the models were also used to inform isolation and quarantine capacity which had a particular emphasis on vulnerable populations including homeless individuals, Native Hawaiian and Other Pacific Island communities, as well as individuals with co-occurring mental illness and substance use challenges. The Hawaii State Department of Health’s Behavioral Health Administration (BHA) was designated to lead isolation and quarantine in August 2020 as HawaiÊ»i experienced its second surge. The BHA was also the sole DOH unit to establish the standalone Temporary Quarantine & Isolation Center specifically for homeless individuals and later for medically needy individuals [17]. As the BHA leadership actively participated in the HiPAM, BHA leadership had sought inputs and guidance from HiPAM models to monitor adequacy of bed capacity for isolation and quarantine real-time case counts. Models from HiPAM were also used to estimate adequacy of shelter capacity for homeless populations.
There was a policy need for a simple benchmark to identify whether there was adequate isolation and quarantine capacity and specifically enough beds procured by the State of Hawaii. With limited time and resources available to conduct a detailed epidemiologic and demographic study, there was a need to identify in a simple manner who might need isolation and quarantine services, for which eligibility was determined in part based on whether an individual was able to safely isolate at home and whether the individual lived in a shared bedroom with someone. In Hawaii, the percentage of the population living in a shared bedroom was identified to be nearly 10%. When applied to the total number of active COVID-19 cases at any given time, this benchmark helped to inform the planning for the total beds procured by the State of Hawaii for isolation and quarantine operational activities in 2020.
Use Case 3: Broad public and media communications
In the Delta and Omicron surges in 2021 in the summer and winter, respectively, HiPAM took a direct public communications strategy to communicate the results of the model and forecast. Rather than use only backdoor communication with senior policymakers and public authorities, HiPAM emphasized direct communications with the media, similar to the weatherman, as well as the release of regular and timely reports sent to all key policymakers and news outlets in the state, supplementing an ongoing web tool with a two-week advance forecast, which had been launched in July of 2020.
By 2021, a locally developed and customized model led by mathematician Chyba et al. became the de facto and well-accepted model in the state, and the other models by University of Basel and IHME were no longer used by 2021 [18]. The Chyba et al. model fulfilled the key considerations for the models including local parameter customizability, local age distribution, use for different policy scenarioing, and being customizable and potentially open-source because it was developed in-house. Developing local in-house mathematical and epidemiologic modeling capacity is extremely challenging and dependent upon availability of scientific experts willing to engage in real-world policy challenges and was buttressed by available funding from the National Science Foundation obtained by Chyba et al. [18].
During the broad public dissemination of the model results to the state in 2021 and 2022, the strategy for communications on modeling was vastly different from the 2020 approach of behind-the-scenes information provided to senior leaders. Nevertheless, this broad public communication strategy also had challenges and risks in terms of the ways in which the modeling results and information was communicated and the kinds of questions and concerns posed by the media, policymakers, and the public. The media and policymakers, for example, repeatedly asked HiPAM representatives challenging questions about the specific policy guidance that should be made based on the modeling results. Yet in order to ensure and maintain the scientific credibility of the HiPAM models, HiPAM repeatedly emphasized its role as a scientific body that focused on high-quality models based on best-available evidence and ever-evolving science. It had to remind the media, the public, and policymakers that policymakers would need to use multiple sources of information, not only from HiPAM, in order to make decisions. In doing so, HiPAM reinforced its role as a scientific body and not as a body making policy recommendations or actions. Maintaining a scientifically neutral and unbiased position was essential for ensuring HiPAM public credibility and recognition.
A second key question repeatedly raised by the public, the media, and policymakers was about the time horizon of the model. There was a longstanding desire for understanding the forecast or projection of COVID-19 well into the future by several months. HiPAM, however, maintained a stance of emphasizing a two-week forecast horizon, and that anything longer than that would be subject to change. Seeing the real-world mistakes of highly public models communicated at the national level, HiPAM made a deliberate choice to focus on a limited time horizon and the ways in which individual and policy actions could easily influence the forecast beyond two weeks. Indeed, HiPAM repeatedly noted that upon release of the forecast, the projection would likely change based on the fact that knowledge and information about the situation would result in changes in individual behavior as well as policy changes and action. This difficulty in communicating the dynamic nature of modeling was challenging throughout the pandemic, even with the seasoned media reporters and regularly engaged policymakers including legislators.
Arguably, the result of the broad public media and communications attention raised public awareness of the COVID-19 modeling writ-large beyond the circles of policymakers and led to broad acceptance and wide recognition and use of the COVID-19 modeling by not only by state agencies but also other health care provider organizations including the local hospital association. The credibility and validity of the HiPAM model work was emphasized primarily through maintaining a neutral stance on any specific policy recommendation, but focusing on the specific technical result or information that the model provided.
In addition, we acknowledged in the Discussion the ways in which other states and countries may have used epidemiologic modeling through the use of a governance entity such as a technical advisory body. In the Discussion, we stated explicitly that we hope that other researchers would be able to do a comparative analysis that examines how modeling was used across multiple states and jurisdictions. We believe this question that the reviewer has posed is extremely valuable, and future researchers can draw upon specific case studies across different jurisdictions and states to better understand how modeling can be used or misused. The Discussion section is now drastically revised and expanded as follows:
- Discussion and Conclusion
Epidemiologic models used for COVID-19 are numerous and complex, requiring content experts to appropriately utilize data and interpret results. The study used a case study approach to reflect on the lessons learned from one jurisdiction in the United States. The historical events reflected in this case study demonstrates the real-world challenges that policymakers face when deciding which model to use, including demonstrating how even with accurate data, utilization of an inappropriate model and/or considerations may lead to inappropriate interpretation of results. COVID-19 models vary by their designed intent and understanding these differences, including their differences in geographic application and their application to specific policy decisions, is necessary for policymakers to better utilize them in making decisions [25–27].
There are several key lessons that can be drawn from this case study documenting the historical application of mathematical and epidemiologic models for key policy decisions. First, COVID-19 modeling in Hawaii benefited from the incorporation of state-specific data which was historically argued to directly result in cost savings from decreased unnecessary spending, particularly in the case of the hospital capacity planning. This model incorporated two of the most important factors that assist local leaders in modeling local issues, age distribution and customization that was specific to Hawaii [9,10]. It also helped to inform isolation and quarantine planning and adequacy of facilities available in order to meet demand and need.
Second, regardless of model selection, it is essential that model outputs be interpreted directionally, not as a forecast of hard, immutable numbers, and reflected in terms of a clearly delineated time horizon. The ways in which the modeling is interpreted and communicated are easily misrepresented. The nearly unlimited variance in combination of factors impacting the spread of COVID-19 means that no model, no matter the level of sophistication, can concurrently take them accurately into account, and the nature of the models, easily influenced by actions of individuals and policies today, make the models dynamic and uncertain rather than static, despite a desire for static and definitive answers. Therefore, numbers produced regarding cases, hospitalizations, deaths, etc. are meant to be understood as a possible scenario should current trends continue forward into the future assuming no change in policy or human behavior.
Third, when the above factors are properly considered and both model outputs – projected trends and subsequent reductions by means of interventions – are combined, it is essential that warnings are heeded and action taken as soon as possible. Based on appropriate interpretation of a model, policymakers can be advised if a policy intervention may avert critical thresholds such as hospital and ICU capacity. With these crucial timings in mind, policy makers can then lean on other models to help more accurately understand how an intervention may impact the R(t) in a given population and, subsequently, what scenarios of intervention combinations and efficacies would result in the elimination of COVID-19 [28]. Hesitation in implementation and/or incompleteness in interventions can have dramatic effects on disease spread, such as the delayed and scattered approach to mask wearing early on in the pandemic [29]. However, failure to grasp model limitations can result in hasty and expensive overreactions, as displayed in the following case study.
Fourth, these models require a firm grasp of epidemiologic concepts. As such, policymakers are advised to involve public health epidemiologists as early as possible to translate modeled outcomes into actionable context. Because of the complexity of models, significant unpredictable impact of human behavior, and the potential for misinterpretation, some have argued that these models do more harm than good. Rather than dismiss the use of models because of their complexity, policymakers should incorporate into their response team a ‘brain trust’ or technical advisory group as early as possible to navigate the difficult policy decisions that can have positive impacts on their constituents and communities. A brain trust is a diverse team which provides input from various areas of expertise (e.g., epidemiology, data science, and mathematics).
The work in Hawaii of using a brain trust may be contextualized to the work in many countries around the world which used COVID-19 modeling and forecasting to inform decision making. In Hawaii, the creation of HiPAM included a broad range of local experts from epidemiology, public health, data science, and mathematics who were able to contribute to modeling and forecasting locally. Other countries such as Ireland, United Kingdom, New Zealand, and several others all had technical advisory groups that provided inputs and information to policymakers who ultimately made the policy calls and decisions. For example, in New Zealand, a COVID-19 technical advisory group comprised medical, public health, and academic advisors, which provided advice to the ministry of health. In Australia, the COVID-19 Expert Database hosted by the Australian Academy of Science provided a mechanism for governments and decision makers to have easier access to expertise in COVID-19. The UK also established a Scientific Advisory Group for Emergencies (SAGE) as the entity responsible for providing scientific advice to UK decision makers while not representing official government policy.
Future research should examine how the creation of these bodies with blended technical expertise and how information informed policymaking and decisions. The composition of these bodies and the extent to which these bodies were linked and connected to decision-making while maintaining neutrality as a scientific body are two key areas for further exploration. We hypothesize that the composition of a body that draws from a wide range of expertise beyond traditional medicine or public health fields can help to bridge challenges of mathematics, epidemiology, and data science. Further, there is a need for communication and practitioners who can help to translate and communicate complex ideas into simple concepts for policymakers and the public. The second area for exploration pertains to linkage to decision makers. Whether forecasting and modeling are sidelined or are integral to decision-making depends on leadership and governance.
There is a large body of research in the field of political science and public administration which examines the ways in which policy decisions and actions are determined and implemented. Political science theories have been applied to understand how political actors make policy actions and political decisions, including Kingdon’s Multiple Streams Model [30] and Reich’s work on political economy [31]. Work by Walt et al. noted that rigorous health policy research methods have much to be desired for understanding the policy process [32]. In particular, a major limitation of this historical case study is its focus on a single jurisdiction — the state of Hawaii – one that lacks a comparison or historical “counterfactual” for what might have happened in the absence of this work on modeling in the state. We implicitly argued that using historical chronology from lived experiences of those engaged in real-world implementation is a research methodology.
This case study also did not examine cases of misappropriation and misuse of models in Hawaii during the pandemic or cases in which the modeling outputs were ignored or otherwise not used for specific policy actions. Determining what constitutes misuse and misappropriation is beyond the scope of this paper, but we acknowledge that the complexity of models makes inappropriate or poor application possible. Future research would be valuable to examine the different ways in which modeling informed or did not inform key policy decisions in multiple states and jurisdictions.
With tremendous uncertainty about a novel disease, the need for thoughtful application of scientific knowledge is ever more pressing. Although the specific use cases and policy window and moment for critical decisions described herein have now passed, the lessons from this case study may be relevant for other jurisdictions seeking to make smarter decisions informed by modeling. The knowledge and experience that was gained through these lived experiences may be applicable for island countries and states with age, ethnic, and other sociodemographic distributions similar to Hawaii. The knowledge and experience from this case study may help to inform other jurisdictions experiencing similar limitations in resources, time, and scientific expertise for COVID-19 modeling in informing policymaking. The institutionalization of knowledge through a brain trust can help policymakers navigate the dazzling array of models to make better informed policy decisions in controlling and mitigating the spread of COVID-19 and other communicable diseases. Existing public health institutions such as those pertaining to health technology assessment or epidemic intelligence may also be suitable for such institutionalization.
Lastly, as an academic research paper, authors need to look up the topic with appropriate theoretical viewpoints. In other words, they need to conduct more academic literature reviews and provide academical implications to the readers who want to understand the meaning of this research in terms of academical research.
Authors’ response: This is a valuable comment, and we have provided a paragraph in the Discussion as follows.
There is a large body of research in the field of political science and public administration which examines the ways in which policy decisions and actions are determined and implemented. Political science theories have been applied to understand how political actors make policy actions and political decisions, including Kingdon’s Multiple Streams Model [30] and Reich’s work on political economy [31]. Work by Walt et al. noted that rigorous health policy research methods have much to be desired for understanding the policy process [32]. In particular, a major limitation of this historical case study is its focus on a single jurisdiction — the state of Hawaii – one that lacks a comparison or historical “counterfactual” for what might have happened in the absence of this work on modeling in the state. We implicitly argued that using historical chronology from lived experiences of those engaged in real-world implementation is a research methodology.
In sum, the topic of this research is meaningful and timely well conducted so that I believe that authors can improve this manuscript well. Thank you.
Authors’ response: Thank you so much for your helpful feedback and comments!
Reviewer 3 Report
The topic of the paper has higher research value, and the discussion of the paper is also relatively sufficient.
However, the research method based on case study has limited the credibility of the conclusions of the paper to some extent. My idea is whether more detailed evidence on the applicability of various models can be provided from the micro data level.
Author Response
Reviewer #3
Comments and Suggestions for Authors
The topic of the paper has higher research value, and the discussion of the paper is also relatively sufficient.
Authors’ response: Thank you so much!
However, the research method based on case study has limited the credibility of the conclusions of the paper to some extent. My idea is whether more detailed evidence on the applicability of various models can be provided from the micro data level.
Authors’ response: We appreciate your consideration of the limitations of a case study, and we note that health policy research and public policy research suffers from this limitation in general — which examines policy analyses of a state or country — are often focused on a particular case study with the absence of a comparison group. The conclusion of the paper is primarily focused on the need for a technical advisory team to help navigate difficult, timely policy decisions using the best data and modeling available in a pressing crisis while acknowledging the issues and limitations of using such tools. Nevertheless, we have taken into consideration the feedback that was provided and appreciate the desire for detailed evidence on the applicability of such models. We have added some clarification as to the context and limitations of modeling in regards to the scope of our paper at the beginning of Section 2:
A “model” refers to a mathematical or logical representation of the biology and epidemiology of disease transmission and its associated processes [4]. To date there are more than 40 COVID-19 models available. In many locations around the world, there was a need for timely public health decisions pressing against the lack of available time, resources, and expertise – and in the case of Hawaii and likely in other locations as well, a severe dearth of epidemiologists in the jurisdiction. Thus, it was not feasible nor practical for policymakers and their technical support teams to comprehensively review all models in order to make decisions. Instead, policymakers were continuously forced to make strategic decisions to select and use tools in order to make the best available information at hand. Nevertheless, even the selection of tools in order to make policy decisions requires technical expertise in order to wade through the science and complexity.
The first part of this case study focuses on the major criteria for selecting a model. Beginning in April of 2020, the HI-EMA tasked a technical team including a lead epidemiologic adviser, a technical analyst, and a working group serving as a technical advisory body to review and use models that would be used to inform a variety of specific policy decisions, described in the second part of this case study. The four models that were ultimately chosen and used for informing Hawaii public authorities for decision making were based on selective review rather than comprehensive review of models. The four models used were the following: the University of Washington Institute for Health Metrics and Evaluation (IHME) model [5], the Imperial College London model [6], the Epidemic Calculator [7], and the University of Basel model [8]. The dimensions for reviewing and selecting these models are described in section 2.2.
Upon review of the documentation and source code, if available, of these models, in 2020, the technical team identified some of the key assumptions of these models. These assumptions were crucial in understanding the limitations or applicability of a given model to a particular jurisdiction, and in this case, the state of Hawaii. These assumptions and limitations are discussed in section 2.3.
2.2. Criteria for model selection
There are several criteria that could be considered for selecting a model. In this case study, the technical team in Hawaii was prompted with questions from policymakers relying on wide media coverage on two models in particular, the University of Washington and Imperial College model. Yet as the technical team discovered, these two models were not completely suitable or customizable for the situation in the local state jurisdiction. The technical team then identified two more models (Epidemic Calculator and University of Basel models) and reviewed these four models based on publicly available documentation (noted in the aforementioned references), and in some cases, data visualizations and source code, along five key dimensions. At the time, the COVID-19 modeling hub had not yet been available in the early part of the pandemic, and thus the models chosen were selective and purposive.
The key dimensions used to select and use these four models were the following: (1) model objective, (2) interactivity and local parameter customizability, (3) age distribution, (4) type of model, and (5) open source (see Table 1). Given limitations of time and resources, the technical team made purposive decisions on which models to consider and use in 2020, and compared and contrasted the models along these dimensions. These dimensions were argued to be relevant for decision-making in Hawaii based on the issues of the assumptions and limitations of the models. While these are not comprehensive of all considerations, they reflect the historical events in the Hawaii case.
We acknowledge the importance of detailed evidence and in the paper do our best to demonstrate this such as through illustrating the issues of the widespread IHME model when considering its use locally in Hawaii. For the reviewer we have elaborated below on the microdata driven decisions regarding the application of the IHME model for urgent policy decision making in the state of Hawaii. As discussed in Section 3.1 the state of Hawaii initially used the widespread IHME model to forecast state-specific estimates of hospitalizations and deaths. In particular, there were urgent policy decisions needed to be made on the need for additional acute care facilities through retrofitting existing hotel rooms or outfitting a convention center. However, even though the IHME model produced state specific numbers it was clear that because the statistical model was fitting using available data from China and Italy the resulting predictions were not lining up with what our local experts knew from our specific non-clinical interventions and unique situation of being an island compared to other states and countries. In the end, while the statistical modeling used by IHME allowed for faster predictions that could extrapolated to the state level, the lack of accounting for the unique geography and other aspects of Hawaii meant the Hawaii specific numbers were off and incongruent and so our technical team searched for a model that allowed for the adjustment of parameters to match our knowledge of local policy decisions.
The challenge of ex post predictive validity compared to real-time decision making and need for other considerations besides predictive validity is now elaborated in section 3 as follows:
While the IHME model influenced policymakers and emergency management leader’’s decisions about imposing public health measures to stop COVID-19 spread (e.g., through closing business and halting travel), it is important to note that the actual death totals from COVID-19 were outside the IHME model’s 95% confidence interval 70% of the time [15]. This fact is notable and particularly relevant given the importance of deaths as a measurement of COVID-19 spread during the early days of the pandemic [16]. Yet predictive validity is an ex post consideration for model selection. Policymakers must make best possible decisions without knowing the future or the predictive validity of any given model. Thus, in the historical case study, the selection of models best used for a given jurisdiction was based on the factors noted in section 2.
Round 2
Reviewer 1 Report
The authors revised the manuscript according to suggestions.
Reviewer 2 Report
Authors' responses are acceptable.